# Progress towards the Elusive Mastitis Vaccines

**DOI:** 10.3390/vaccines10020296

**Published:** 2022-02-15

**Authors:** Pascal Rainard, Florence B. Gilbert, Rodrigo P. Martins, Pierre Germon, Gilles Foucras

**Affiliations:** 1ISP, INRAE, Université de Tours, UMR1282, 37380 Nouzilly, France; florence.gilbert@inrae.fr (F.B.G.); rodrigo.prado-martins@inrae.fr (R.P.M.); pierre.germon@inrae.fr (P.G.); 2IHAP, Université de Toulouse, INRAE, ENVT, 31076 Toulouse, France

**Keywords:** mastitis, vaccine, cattle, humoral immunity, cell-mediated immunity

## Abstract

Mastitis is a major problem in dairy farming. Vaccine prevention of mammary bacterial infections is of particular interest in helping to deal with this issue, all the more so as antibacterial drug inputs in dairy farms must be reduced. Unfortunately, the effectiveness of current vaccines is not satisfactory. In this review, we examine the possible reasons for the current shortcomings of mastitis vaccines. Some reasons stem from the peculiarities of the mammary gland immunobiology, others from the pathogens adapted to the mammary gland niche. Infection does not induce sterilizing protection, and recurrence is common. Efficacious vaccines will have to elicit immune mechanisms different from and more effective than those induced by infection. We propose focusing our research on a few points pertaining to either the current immune knowledge or vaccinology approaches to get out of the current deadlock. A possible solution is to focus on the contribution of cell-mediated immunity to udder protection based on the interactions of T cells with the mammary epithelium. On the vaccinology side, studies on the orientation of the immune response by adjuvants, the route of vaccine administration and the delivery systems are among the keys to success.

## 1. Introduction: Are Efficacious Mastitis Vaccines Feasible?

Bacterial infections of the mammary gland (MG) are a major issue in dairy cows. It is the most common reason for antimicrobial use in dairy farms [1]. Efficacious vaccines would be very helpful, but unfortunately, currently they are not available, despite decades of research [2]. The question that arises is: can we design new approaches to achieve protection against MG infection through vaccination? Some scientists have asserted that the notion of rational vaccine design is a bogus claim in the current state of knowledge of the immune system and host-pathogen interactions [3]: the rational design of vaccines would be a notion developed by smooth talkers. In the mastitis field, a similar position was stated in the late 1970s: “Currently the attitude is of exasperated optimism” [4], and the possibility of developing a vaccine against *Staphylococcus aureus* mastitis has been doubted. Indeed, if we confront the glut of vaccine trials to the results in terms of efficacious mastitis vaccines presently in use, the expression “trial and error” takes on its full meaning [2]. It must be admitted that none of the mastitis vaccines currently on the market are satisfactory [2]. Of course, it depends on the height where the bar is set: most mammary infections are persistent infections, we cannot be satisfied with limiting the number and severity of clinical episodes. We must seek to obtain a bacteriological cure (Figure 1). To help improve such an unfortunate state of affairs, the idea that vaccine development would benefit from the injection of more immunology knowledge into empirical vaccinology has been put forward [5]. Several theories have been proposed to promote diverse approaches to tackle pathogens refractory to vaccine control, i.e., reverse vaccinology or systems vaccinology [6,7]. Would new advances in immunology and vaccinology make the possible advent of more efficacious vaccines to prevent or even cure mastitis? Several reasons render the development of efficacious mastitis vaccines difficult. Their identification and analysis would help us to make an objective assessment of the issue and could raise hopes that a number of prospective solutions can be envisaged. Among the impediments opposed to mastitis vaccine development, some were identified long ago and are well documented. They are related to the MG physiology and immunology, or the type and virulence of causing bacteria. Other obstacles are more speculative, such as the possibility of eliciting sterilizing immunity in the MG against commensal bacteria that do not usually induce this response following infection, but their identification may offer new perspectives. Finally, practical reasons discourage pharmaceutical companies from embarking on research and development of vaccines against bovine mastitis. A preliminary answer to these concerns is improving efficiency.

## 2. Possible Reasons for the Current Mastitis Vaccine Shortcomings

### 2.1. Obstacles to Effective Vaccination That Are Peculiar to MG Immunobiology and the Diversity of Pathogens

A number of obstacles have been identified, such as the paucity of immune agents in milk, the inhibitory effects of milk components (fat and casein) on phagocyte functions, the vast surface of secretory mammary epithelium requiring immunological surveillance, and the excellent growth medium provided by milk for many bacterial pathogens [8]. In the udder, immune responses are fraught with limitations with regard to phagocytic defense mechanisms [9]. Phagocytosis by neutrophils recruited in the MG is arguably the major defense mechanism against most mastitis-causing bacteria, and vaccination can improve opsonophagocytosis [10]. Unfortunately, the MG environment is not favorable to phagocytosis. Milk interferes with phagocytosis and kills efficacy by neutrophils that ingest fat globules and casein micelles [11]. Low oxygen tension in milk is another impediment reducing the bactericidal activity of neutrophils [12], and the milk matrix is a quencher of reactive oxygen species. Opsonins are also limiting factors for the phagocytosis of certain encapsulated bacteria. The deposition of the opsonic Complement fragments C3b/C3bi usually occurs in milk, but complement is in short supply until inflammation provides supplements through the exudation of plasma [13]. Antibody concentrations are low, in particular in the IgM and IgG_2_ isotypes that are opsonic for neutrophils [9]. However, natural antibodies are present in milk in sufficient quantities to opsonize the most common pathogens, and the exudation of plasma enhances this supply through inflammation.

A peculiarity of the MG niche for bacteria is that its lumen is filled with milk during the lactation period. Milk is a nutrient-rich medium for bacteria equipped with enzymes, enabling the degradation and use of lactose (a general attribute of mastitis-causing bacterial isolates) and casein [14]. Others have developed strategies to overcome their handicap, such as the plasminogen activator of *Streptococcus uberis* [15]. The antibacterial systems once called lactenins [16] (lactoferrin, lactoperoxidase, xanthine oxidase, lysozyme, antimicrobial peptides) may lack some cofactors to be fully active or have a limited spectrum of activity against bacteria that have evolved countermeasures during their co-evolution with their bovine host [17,18]. Milk also contains molecules that participate in recognition of bacteria and the triggering of the inflammatory response [19]. However, host defense-related milk proteins are diluted in large volumes of milk and drained at each milking, so their concentrations are often insufficient. In principle, antibodies could prevent the adhesion of bacteria to the mammary epithelium. In most mucosal epithelia, the inhibition of adherence is affected by secretory IgA [20]. Of note, IgA concentration is low in the MG of ruminants, and there is no mucus to anchor secretory IgA at the surface of the epithelium lining at active concentrations [21]. In dry MGs, dilution and quenching are less of a problem, and indeed clinical mastitis cases are not frequent, except during the first two weeks of involution following the cessation of milking. However, subclinical infections may persist, waiting for favorable local conditions to develop in full mastitis in the peripartum period [22].

Even though we consider only the main bacteria that cause the bulk of mastitis cases, there is still a wide range of microorganisms that can cause mastitis. In a preceding review of the mastitis vaccines that aim to cope with staphylococcal, streptococcal, and coliform bacteria, one common theme was the diversity of surface antigens displayed by the pathogens [2]. The expression of the virulence-associated factors can also be very variable. Antigens shared by all or sizeable proportions of isolates have been identified, but the efficiency of the immune response they elicit has not convincingly been established. An exemplar is the diversity of mastitis-associated *Escherichia coli* O serotypes [23,24,25] and the polysaccharide serotypes and surface proteins of *Streptococcus agalactiae* [26]. As regards *S. aureus*, the variability of the surface antigens and of their expression makes it difficult to identify consensus antigens [27]. Ways to find a parry to this complexity have been to look for the antigens that elicit antibodies during infection and to find antigens shared by most mastitis-associated strains with minimal strain-specific variations. It is worth noting that there is no guarantee of proper orientation of the immune response and protective activity associated with such antigens.

### 2.2. The Issue of Protection Induced by MG Infections

A strong objection that can be made against the possibility of developing efficacious vaccines against mastitis is that natural infection of the MG does not improve protection. Most efficacious vaccines have been developed against infectious diseases that induce complete and long-lived protection, such as yellow fever, measles, mumps, plague, tetanus, or rabies, mainly through the induction of antibodies [28]. There are arguments to support that mastitis does not induce protection through immune memory. An udder quarter infected with *S. aureus* and cleared by treatment after several weeks, i.e., time sufficient for adaptive immunity to develop as shown by titer increases of specific Abs can be infected with the same strain, with the same small inoculum size [29]. There is no obvious modification in the way the MG deals with the pathogen. Subclinical mastitis may not induce protection because the intensity of the antigenic stimulus is too low (both in terms of antigen and co-stimulus). That is suggested by the low antibody titers induced by subclinical infections compared to the higher titers induced by clinical episodes, as exemplified by *S. aureus* infection in goats [30]. It has been proposed that the antigenic effect of chronic *S. aureus* infection on the memory type of immune response in the bovine mammary gland is minimal. The persistence of *S. aureus* infection may result, in part, from suboptimal stimulation or immunosuppression of the mammary immune system [8,31]. Experimental infection of the MG with a capsulated strain of *E. coli* induced opsonic Ab in milk, but the cows often became severely ill when challenged with the same strain in the subsequent lactation [32,33]. It has been reported that spontaneous recovery from *S. uberis* infections does not preclude recurrence with related or different strains [34]. Circumstantial evidence that previous encounters with *E. coli* do not improve clinical signs is provided by older cows, which are more susceptible to severe coliform mastitis (strong parity effect) than primiparous cows [35]. This observation holds for other MG pathogens, leading to the conclusion that “clinical mastitis does not generally protect against a recurrent case” [36].

However, it is inexact that infection never changes the capacity of the infected gland to react to a new encounter with the same bacteria. It has been reported that a cow that became severely ill when experimentally infected with a capsulated strain of *E. coli* (B117) acquired opsonic activity towards this strain in her milk and rapidly cleared the infection when challenged with the same strain 150 days later [37]. A significant reduction in the clinical mastitis upon intramammary challenge with *S. uberis* 0140J was obtained on the second challenge carried out when local inflammation was resolved [38]. Curiously, this protection was not shared by all the quarters of the partially protected cows, as most of them developed clinical mastitis in one of the two re-challenged quarters. In these examples, phagocytosis-resistant strains were used, and the protection was likely to be strain-specific, since with live *S. uberis* vaccines, the protection was essentially against the homologous vaccine strain [39]. In the same vein, udder quarters infected with *Mycoplasma dispar* were then cured, but not the uninoculated quarters of the same udders; they reacted more strongly (neutrophil influx and plasma exudation) and quickly eliminated the bacteria upon reinfection [40]. Infection with a capsulated *E. coli* strain augmented milk opsonic activity only for strains of the same capsular type [32]. Cows with a history of *S. aureus* MG infection were reported to react with an increased response to intradermal stimulation with killed staphylococci [41]. However, the same authors pointed out the difficulty of increasing the natural antibody levels by vaccination with whole bacteria.

It can be concluded that the natural infection of the MG tends to induce a narrow and moderately effective immune response, mainly by reducing the severity of the disease induced by virulent strains and by marginally increasing the capacity of the gland to heal it. This is reminiscent of the results obtained with live vaccines, attenuated or not. Live vaccines are supposed to be more efficacious than killed vaccines (bacterins) because they express “in vivo antigens”, i.e., antigens that are expressed in the host but not in culture (or in low amount) [42]. They also tend to elicit a different immune response: live *S. aureus* given subcutaneously induce mainly an IgG_2_ antibody response and an amplified influx of neutrophils into the MG, whereas killed *S. aureus* given with oily adjuvants stimulate the production of predominantly IgG_1_ antibodies and no change in neutrophil recruitment at the onset of an experimental infectious challenge [43]. Likely, a different stimulation of antigen-presenting cells by live versus killed bacteria results in different polarization of the T cell response.

In conclusion, it is clear that natural IMI itself does not elicit complete protection against reinfection, so that any vaccine would have to improve on the natural immune response. An example is provided by the antibody response to *Streptococcus pyogenes* inhibitor of complement SIC: natural infection does not elicit neutralizing antibodies, whereas vaccination with the recombinant protein does [44]. Accordingly, the new vaccines would protect against infectious diseases by eliciting “unnatural immunity”; that is, immune responses that are of a different nature and directed to different antigens from those elicited by natural infections [5]. In other words, believing that the immune response induced by infection can be harnessed for protection against pathogens that do not induce good protection might be more ingenuousness than ingenuity.

### 2.3. Is Mastitis an Infection without Virulence Factors?

It is worth noting that most mammary pathogens are commensal bacteria. A commensal microorganism lives with its host without causing disease. Among commensal bacteria, there are opportunistic pathogens that possess virulence factors that allow them to damage the host. In the particular case of the MG niche, bacteria need some fitness attributes to grow within mammary secretions (i.e., iron acquisition system, or enzymes degrading casein or lactose) or to gain and maintain a foothold in the MG (surface adhesins), along with protective attributes that allow bacteria to passively resist host defenses (e.g., capsule or slime production). However, should these bacteria possess major factors determining virulence for the MG? To be more specific, we will distinguish passive virulence, such as capsules or slime, from active virulence factors, such as toxins or enzymes. This is more than a semantic question because virulence factors are handles for the immune system, offering targets for specific protective immunity. By far, the most frequent MG infections are by coagulase-negative staphylococci (CoNS). Arguably, CoNS are well-adapted parasites of the MG: if we consider the capacity of bacteria to establish long-lived intramammary infections (persistent mastitis) and equate this capacity to the adaptation to the MG niche, then the less aggressive the strain, the more adapted it is. CoNS are well equipped to produce biofilms. Biofilm formation is a form of passive resistance to host defenses (such as phagocytosis), a growth behavior offering protective advantages without triggering inflammatory responses [45]. Few, if any, CoNS possess a leukotoxic activity [46]. Most CNS species and strains can cause long-lasting IMI without symptoms other than moderately increased somatic cell count (SCC) and sometimes without noticeably decreasing milk production [47]. Hence, CoNS colonization of the MG is, most of the time, an infection without disease in cows (small dairy ruminants differ in this respect), and many CoNS could be considered mastitis-associated pathogens without virulence. However, CoNS are opportunistic pathogens, which can be a cause of milk yield reduction [48], and epidemiological surveys have shown that CoNS can be responsible for a sizeable proportion of clinical mastitis cases, some of which can be very severe when immune defenses are compromised [49]. The CoNS example shows that bacteria do not need to possess aggressive virulence factors to colonize the MG but also to cause severe disease if the immune system lowers its guard.

By contrast, mastitis-associated *S. aureus* strains possess an armamentarium of virulence factors [50]. However, in the cow, a good proportion of IMI are subclinical, and chronicity has been associated with stealthy behaviors best exemplified by the small colony variant (SCV) intracellular form and its reduced expression of virulence factors [51,52]. Similar examples of reduced virulence can be found with other mastitis-associated pathogens. Strains of *S. agalactiae* of human origin induced more severe MG infections than did strains of bovine origin, yet the immune response eliminated the former, whereas the latter persisted, suggesting that adaptation to the MG is accompanied by virulence reduction [53]. Other examples are provided by *S. uberis* and *E. coli*. Mastitis-associated *S. uberis* strains are not aggressive to mammary epithelial cells (MEC) and tend to evade recognition by these cells [54]. There is no gene described as an aggressive virulence attribute associated with *S. uberis* strains able to induce clinical mastitis [55]. Mastitis-associated *E. coli* strains rarely possess the virulence attributes of other pathogenic *E. coli*, and they are mainly of the commensal phylogenetic groups [56,57]. Their pathogenicity for the MG is essentially linked to the LPS endotoxin, which is somewhat of a misnomer as LPS is cytotoxic only at very high concentrations and mainly owes its toxicity to the exacerbated inflammation it triggers.

If we consider that virulence factors stricto sensu are aggressive substances that target defined cell types (e.g., leukotoxins, hemolysins) and specifically counter immune defenses (e.g., inhibitors of complement or chemokines), it appears that bacteria do not need these attributes to colonize the MG. On the contrary, a loss of these aggressive attributes may accompany adaptation to the MG, as suggested for *S. aureus* [58,59]. The MG, more specifically, the lumen of the lactating MG, presents an intrinsically low level of immune defense, and is prone to “infection without virulence”; that is, infection without required aggressive virulence factors. One implication is that there is no major target for vaccine design, a target that would elicit a decisive protective response on its own, but at best a multiplicity of minor targets related to bacterial fitness rather than to virulence.

### 2.4. Co-Evolution of the MG and Mastitis-Causing Pathogens

The most prevalent mastitis pathogens are part of the normal microbiota of dairy ruminants. This is why aseptic precautions must be taken when sampling the MG because the mere isolation of *S. aureus*, *E. coli,* or *S. uberis* from milk samples does not mean that the bacteria come from an IMI rather than from contamination or colonization of the teat canal. The long-lived close association of these bacteria with their host gave them ample time to evolve resistance mechanisms to thwart most of the host immune defenses. It can be put forward that the co-evolution of the mammary gland with mastitis-causing pathogens led to a status quo, best illustrated by the classical chronic subclinical mastitis induced in cows by *S. aureus*, *S. uberis*, or CoNS. Chronic mastitis allowing long-term shedding of bacteria in milk favors the transmission of the pathogens to other cows. Subclinical infection has allowed for the maintenance of milk secretion and offspring survival under natural conditions (wildlife before domestication).

Infection by Gram-positive mastitis pathogens induces a spontaneous immunity that rather effectively controls bacterial proliferation but does not lead to bacterial clearance. By enhancing this naturally induced immunity, the result might be an exacerbation of ineffective inflammation but not a bacteriological cure. The immune response can be manipulated by the pathogen, an example being *S. aureus* superantigens that dysregulate and mislead the immune response into inefficiency [60]. Admitting that the objective of vaccination is to induce “unnatural immunity” has implications for the direction vaccine research could take. For example, immune responses induced by IMI could be regarded with suspicion as they may contribute to the status quo. Approaches such as the serological proteome analysis (SERPA) rest on the idea that antibodies induced by the infection are potentially beneficial. This can be doubted in the case of infections that do not induce protection that prevents recurrence. The natural immune response may contribute to the persistence of infection, albeit at a subclinical level, through diversion from more effective immune responses. The pathogen may drive the immune response in a direction allowing it to persist, as exemplified with human T cells by *S. aureus*, which induced Th17 cells producing IL-17 and IL-10, whereas *Candida albicans* induced Th17 cells producing IL-17 and IFN-γ [61]. The case of *E. coli* may be somewhat different. These bacteria normally reside in the intestine, not in the MG thus they co-evolved with their host in this digestive niche, developing fitness traits, some of which also increase their fitness to other niches [62]. In this case, it is the MG that evolved to respond to opportunist *E. coli* in a way that does not durably compromise its secretory function (the capacity to recover milk secretion shortly after an abrupt drop in milk secretion in response to *E. coli* MG infection is spectacular). As noted, many MG infections are caused by bacteria that are members of the microbiota colonizing organs other than the MG (intestine, skin, upper airways). The co-evolution of pathogens with their hosts on a commensal mode is likely to induce two reciprocally adaptive responses. On the one hand, the pathogen tends to bias the host immune response towards tolerance. An illustration is provided by the interaction of *S. aureus* with MEC, where the NFκB pathway would not be stimulated and invasion favored [63], or the immune response dampened by the induction of TGF-β in MEC [64]. On the other hand, innate and adaptive immune responses to commensals (and among them opportunistic pathogens) are calibrated to protect tissue homeostasis [65]. In the concept of homeostatic immunity, regulatory T cells play an important role. We know little of the occurrence of regulatory T cells (CD4+ Treg, CD8+, or γδ T cells) in milk and mammary tissue and if they control the immune response to opportunistic pathogens that are members of the cow microbiota. However, we know that *S. aureus* can induce regulatory T cells in humans, mice, and cows [66,67,68].

### 2.5. Hypothesis: The Immune Response to Mastitis-Causing Bacteria Is Biased towards Tolerance or Acceptance

Repeated or permanent contact with commensal bacteria is likely to induce an immune response that contributes to the long-term balance of bacteria–host interactions. Adult cows are not naïve to these bacteria, and, consequently, they are likely to possess memory T cells that contribute to the status quo, establishing a modus vivendi with the bacteria that are present in the digestive or cutaneous microbiota. In the intestine, different populations of lymphocytes arise from the interaction with the microbiota. Among them, lymphoid cells secreting pro-inflammatory cytokines, such as IL-17A and IL-17F, are balanced by regulatory cells that maintain the inflammation at a low grade, and the homeostasis of the intestine can be considered as a controlled inflammation [69]. Type 3 immunity, which is mediated by immune cells that secrete IL-17A, IL-17F, and IL-22, is likely to play an important part in the defense of the MG against infections [70]. If not controlled, this immunity may cause tissue damage through exacerbated inflammation. According to the Danger theory of immunity, persistent contact with skin, mouth, upper airways, or gut commensal bacteria without cellular damage may tolerize antigen-specific T lymphocytes [71]. We can hypothesize that regulatory T cells specific to commensals are present in the MG, which would blunt the immune response to these bacteria. This dampening could make sense because overreacting to these commensals could be harmful, displacing the balance of accepted host/pathogen cohabitation towards conflicting inflammation, jeopardizing the secretion function of the MG. This mode of handling of MG infection would have been positively selected over the age-long co-evolution of the MG with commensal bacteria because of its overall positive effect on the survival of the suckling offspring. However, these commensals are opportunistic pathogens when the occasion arises; that is, whenever and wherever the immune defenses are depressed. The resulting blunted immune response would hamper the control of infection, both acute coliform mastitis and subacute streptococcal or staphylococcal infections. The state of hyporesponsiveness does not necessarily result from the occurrence of regulatory T lymphocytes, although this possibility deserves investigation. In the mouse MG, the depletion of Treg cells during an *E. coli* infection prevented the early production of IL-10 and improved the clearance of bacteria [72]. However, Treg cells can also assist in the production of IL-17 and IL-22 by Th17 cells and promote pathogen clearance [73]. It will thus be necessary to deepen our understanding of Treg (or other regulatory T cells) and Th17 cell interplay in the MG. Whatever the mechanisms of regulation, eliciting “unnatural immunity” would disrupt the cohabitation balance and cause an exacerbated but transient immune response, acceptable only if it results in bacteriological cure without sizable tissue damage.

We do not know whether there is a real trend for active tolerance in the MG or only a mere passive acceptance of colonization by a mild pathogen. Studies of *S. aureus* infections do not provide a clear answer. The regulation and counter-regulation of the adaptive immune response to *S. aureus* occur during infections in certain mouse models [74]: immunity develops to maintain a pacific coexistence between *S. aureus* and its host. In return, the adaptive immune response induced by infection does not lead to sterilizing immunity. Rather, the end effect seems to be the control of *S. aureus*, and the establishment of the host–pathogen equilibrium in case of new bacterial invasion [74]. Chronic infection by *S. aureus* (mouse model) is often associated with the development of abscesses under the control of CD4 T cells secreting IL-17. This form of bacterial containment leads to the persistence of bacteria within the host and maintains low to medium grade inflammation in the tissues. Chronic *S. aureus* infections may also induce a state of Ag-specific T lymphocyte anergy and some non-specific immune tolerance [75]. In this line of thought, contrary to *E. coli*, *S. aureus* can decrease the production of IL-32 by bovine MEC through the secretion of phenol-soluble modulins, which could impair the immune response by interfering with dendritic cell maturation [76]. Regulatory CD8 T cells able to suppress CD4 T cells have been found in the milk of cows with *S. aureus* IMI [77]. Systemic physiological or pathological events could modify the inflammation-tolerance balance. A well-known example is the immunity perturbation during the periparturient period [78], which tips the balance towards susceptibility to MG infection. Vaccine-induced modification of the balance between pro-inflammatory and adaptive regulatory T cells is likely to be a crucial issue for safety and efficacy.

We have seen that there are a number of possible obstacles to mastitis vaccine development (summarized in Table 1). We will now propose a few research leads to get out of the current deadlock.

## 3. What Can Be Proposed to Get out of the Mastitis Vaccine Predicament?

The literature is replete with articles reporting vaccine trials and attempts to use diverse antigens to induce protection against mastitis (illustrative examples in Table 2). Most research has been devoted to antigens and their route of administration, with the objective of inducing antibody responses. Comparatively, little work has been devoted to other relevant immune responses. Vaccine shortcomings may result, in part, from an overreliance on antibody-mediated protective response. Moreover, vaccinology research outweighs immunology research, and this imbalance may have retarded progress [79]. Vaccinology is microbe-centered and immunology host-centered [5]. Most research recently has focused on the search for bacterial protective antigens. Research on cell-mediated immunity in the MG has been neglected since the late 1990s, whereas, in the meantime, new tools and concepts have emerged in the last twenty years. Identifying major protective immune responses that vaccination can induce in the MG is a prerequisite to rational vaccine design. Validating reliable correlates or surrogates of protection would result from this new knowledge, which would be a great help to vaccine development. Given the complexity of the immunobiology of the MG and its specificities, as well as the pathogenesis of MG infections, it is necessary to narrow in on the most relevant defenses and to prioritize vaccine objectives. Accordingly, vaccinology research can be oriented, focusing on immunogenic antigens, adjuvants, route of administration, and delivery systems and timing, to orient the immune response towards protection. Much empiricism will remain in vaccine development, but at least the experimenters will be able to learn from their vaccine failures or shortcomings.

### 3.1. Progress in Immunology

#### 3.1.1. Filling the Main Knowledge Gaps in Cell-Mediated Immunity in the MG

Surprisingly, little is known about lymphocyte trafficking to the MG of ruminants, both in terms of vascular addressins and complementary structures on lymphocyte membranes, the homing receptors. Apparently, mammary lymphocytes in sheep and cattle have migration properties different from those of non-ruminant species. Importantly, it appears that the mammary immune system is not closely linked to the intestinal immune system and that MG lymphocytes originate from peripheral rather than mucosal sites [96,97]. This concurs with other observations, such as the lack of mucus secretion, limited production of secretory IgA, and the dearth of subepithelial lymphoid formations, to reach the conclusion that the MG of ruminants is not a classical mucosal organ. However, a local immune response can be elicited by intramammary immunization during the dry period [98,99]. These considerations have implications for the possible routes of vaccine administration potentially effective in inducing lymphocyte seeding of the MG. Systemic immunization routes can induce local cell-mediated immunity provided peripheral tissue is seeded with memory T cells [100]. This likely occurs in the cow as it is possible to elicit antigen-specific neutrophilic inflammation by injecting a soluble antigen into the MG lumen after subcutaneous immunization with that antigen [101,102]. Such a reaction indicates that resident memory lymphocytes can be induced to populate the MG by systemic immunization. That the reaction correlates with the induction of antigen-specific CD4 T cells producing IL-17A and IFN-γ [103] suggests that Th17 cells are involved. Bovine Th17 cells have been shown to produce these two cytokines [104]. Local (intramammary) booster immunization following systemic priming proved to be an efficacious inducer of resident CD4 T cells and type 3 immunity [105]. Regrettably, the homing determinants of vaccine-induced lymphocytes prone to migrate to the MG remain undetermined. Moreover, mammary T cells are poorly characterized both phenotypically and functionally. Even milk lymphocytes have seldom been characterized beyond the expression of CD4 and CD8 markers [106,107], and markers are generally considered to be associated with the memory phenotype [108].

The specific contribution of various cell types susceptible to affect the defense of the MG against infections, such as CD4 and CD8 αβ T lymphocytes, γδ T lymphocytes, innate lymphoid cells, and NK cells, remains to be delineated. This represents a substantial task. Potent new tools are available for fine analysis of cellular and molecular immunobiology of the MG [109,110], but unfortunately, the meager funding of ruminant basic immunology hinders their use.

#### 3.1.2. Promote T Cell Immunity in the Framework of MG Immunobiology

Mastitis is essentially a duct disease; a statement substantiated for *S. aureus*, *E. coli*, and *S. uberis* IMI [10,111,112]. This means that essential defenses must be efficient at the level of the epithelial barrier. Even though live bacteria can be found in the draining lymph nodes, mastitis is usually not an invasive disease, and bacteremia is rare, even in the most severe cases [113]. This suggests that the immune responses are adapted to prevent secondary infection foci (metastatic infections) and do not require improvement. Protection against mastitis thus equates essentially to the protection of the MG epithelium. In addition, the lumen of the MG must be rendered inhospitable for bacteria. This is mainly the task of neutrophils that have to be recruited swiftly and en masse to control the exponential growth of bacteria [114]. The principal immune response in charge of both epithelium defense against extracellular bacteria and neutrophilic inflammation is type 3 immunity [115]. Type 3 immunity has two arms, one innate, the other adaptive, which cooperate closely. Type 3 immunity is mediated by lymphoid cells, including adaptive CD4+ Th17, CD8+ Tc17 cells, and innate lymphoid cells (ILC3), producing the signature cytokines IL-17A, IL-17F, and IL-22, which induce epithelial cell antimicrobial response and neutrophilic inflammation.

The production of the signature cytokines IL-17A, IL-17F, and IL-22 in milk over the course of *S. uberis*, *E. coli*, and *S. aureus* mastitis suggest that innate type 3 immunity is part of the natural immunity to mastitis [116,117,118,119,120,121,122,123]. With respect to adaptive immunity, local immunization induces resident type 3 immunity-related CD4 T cells in the mammary tissue [105]. Antigen-presenting cells, such as the MHC II+ ductal macrophages, are ideally positioned in the bilayer epithelium to sample antigens in the lumen and present them to subepithelial lymphocytes [124,125]. We have mentioned that the natural immunity elicited by infection does not afford complete protection. Intramammary infusion of IL-17A in the mammary gland of mice infected with *E. coli* has helped to control the infection, suggesting that increasing the local production of this cytokine early in the course of infection could be beneficial [72]. Vaccination will have to add an adaptive component to the innate and to the likely pre-existing anamnestic immune responses by mobilizing the appropriate effectors (Figure 2). Th17 cells are particularly active in the control of opportunistic pathogens that typically co-exist with the host as part of the commensal microbiota [126]. Encouraging examples of Th17-dependent protection have been obtained in mouse models of infection with *S. aureus* antigens [127,128]. Theoretical, observational, and experimental evidence support a role for type 3 immunity in the defense of the MG against infections [70]. Interestingly, adaptive type 3 immunity is able to synergize with innate immunity in the MG [129]. The optimal conditions of the induction of this type of response and the precise characterization of its effector cells is a promising research area.

Another pathogenic mechanism that is likely to contribute to the persistence of IMI is the invasion of MEC and possibly of macrophages by bacteria that survive within those cells by adopting stealthy behavior. A possible immune response able to deal with this evasion mechanism is mediated by cytotoxic CD8+ T cells. These cells could flush bacteria out of infected cells and make them accessible to phagocytic cells or directly kill the bacteria, as shown for *S. uberis* [130]. This approach has been advocated for *S. uberis* mastitis [131] but could well apply to *S. aureus* and persistent *E. coli* IMI. Bacteria-specific CD8 T cells can be induced by the cross-presentation of antigens. Cross-presentation is a phenomenon occurring mainly in dendritic cells by which antigens picked up by endocytosis are processed and presented by major histocompatibility complex class I (MHC I) molecules to CD8 T cells [132]. Cytotoxic CD8 T cells detect target cells in a TCR and MHC I-dependent manner that requires cell-to-cell contact [133]. There was no clue that infected MEC present bacterial antigens associated with MHC I molecules. However, immunohistochemistry analysis has shown that CD8+ T cells are frequently seen in close apposition with the mammary epithelium [134,135]. Another way to induce T cell cytotoxicity would be through the presentation of bacterial peptides by MHC II molecules since cytotoxic T cells may originate from the reprogramming of CD4 T cells [136]. These intriguing possibilities offer new research leads.

An advantage of T cell immunity is that the T cell peptide epitopes are often conserved and shared between bacterial strains, thus conferring wider protection than does antibodies [137,138]. Another potential asset is that CD8 T cell responses are particularly suited to deal with chronic infectious diseases [139]. However, inducing T cell responses in the proper polarization is not an easy task [140]. Using live attenuated bacteria with an invasive phenotype to deliver antigens intracellularly is a possibility [141]. T cell-based vaccines are promising, but several hurdles will have to be cleared. Simple correlates of protection are lacking, and the polarization of the response is crucial but difficult to achieve.

#### 3.1.3. Define Valid Correlates of MG Protection

In the vaccine design context, the broad definition of an immune correlate of protection is a vaccine-induced immune response that predicts protection from infection or disease. A reliable general correlate of protection has to be demonstrated in different settings and populations [142,143]. Ideally, a good correlate supposes a link of causality of the immunological marker with protection, but this is not necessarily the case. According to these requirements, it is clear that there is no documented correlate of protection for mastitis vaccines. Notwithstanding, several putative correlates of protection can be considered. Functional signatures can be used as correlates of protection. Initially, antibodies neutralizing viruses or toxins were considered the main correlates of protection. This belief was supported by the observation that most of the early successful vaccines protected through antibodies. This is the case for vaccines against tetanus, diphtheria, rabies, and yellow fever [144]. With the development of immunochemistry, it became relatively easy to measure antibody responses, and thresholds above which antibody titers were considered protective could be set up. This approach was tentatively used to evaluate mastitis vaccines. Changes in antibody titers often serve as the gold standard to assess the response to the administered vaccine. Failure of a vaccine to increase titers is often regarded as a vaccination failure. However, predicting direct correlations between antibody titers and clinical outcomes in the face of natural exposure to the pathogen is tenuous, as exemplified with J5 vaccines [145]. Generally, such correlations are low and not significant. It is worth noting that protection by antibodies depends on their effector activities, such as the neutralization of virulence or toxin, blocking of adhesion, activation of complement, promotion of opsonophagocytosis, besides their specificity and affinity. Consequently, ELISA titers may be misleading because ELISA also detects low-affinity antibodies, and low affinity may be insufficient for neutralization. Moreover, the recognized epitopes may not be the right targets of effector activity. They can also be shielded by a bacterial capsule or other surface polysaccharides, as are the outer membrane proteins of *E. coli* mastitis isolates, which are beyond the reach of the antibodies induced by the rough *E. coli* J5 [146].

The phagocytic killing of invading bacteria is an established major defense mechanism of the MG, and opsonizing antibodies are necessary for optimal phagocytosis. However, there is scant evidence that increasing the concentrations of opsonins through vaccination is a requisite to induce protection. For example, following immunization with *E. coli*, titers of antibodies in the opsonic immunoglobulin isotypes increased, but the ex vivo phagocytic killing efficiency in inflammatory milk was not improved [117]. We have seen that dairy ruminants possess natural antibodies, mainly of the IgM isotype, that are opsonic and with concentration and spectrum sufficient to opsonize most mastitis-associated bacteria. High titers of natural antibodies in the IgM isotype have been reported to be associated with a lower risk of clinical mastitis [147]. The adhesion of bacteria to MEC is supposed to facilitate infection of the MG thus antibodies inhibiting adherence may be useful, although not decisive. The prevention of MEC invasion might reduce the likelihood of the establishment of a persistent infection. As with opsonins and antitoxins, the difficulty with anti-adhesins is to maintain active concentrations in milk during lactation.

Measuring the functional response of T cells as a correlate of protection has attracted a growing interest lately. Many human infectious diseases that are top priorities for vaccine development, such as HIV, TB, and malaria, require strong T cell responses for protection. However, establishing the functional signature of T cell responses as a correlate of vaccine efficacy is not straightforward and, so far, no vaccine protects through T cell immunity only [140,144]. In the context of vaccination to bovine infections caused by *Mycobacterium* species, indirect (production of IFN-γ upon antigen stimulation of PBMC) or direct (elicitation of cytotoxic CD8+ T cells) evaluation of cell-mediated immunity is performed [148,149]. In the mastitis vaccine context, T cell-mediated responses are seldom reported. They may be more relevant to protection and could be more durable than the antibody titers to commensal bacteria, which usually dwindle in a few months. There is a need to find immune correlates of cell-mediated immunity to characterize the responses to mastitis vaccines.

### 3.2. Progress in Vaccinology

An improved understanding of the MG immunity will be most useful to the design of efficacious vaccines, but this achievement will not be possible without the recourse to the best tools and approaches provided by vaccinology. It is beyond the scope of this perspective article to systematically review all elements of vaccinology relevant to the design of a mastitis vaccine. We will only focus on those issues that seem to deserve special attention with regard to the inherent characteristics of MG infections.

#### 3.2.1. Capitalize on Systems Vaccinology

Systems biology aims to take into account the complex interactions between all parts of a biological system, with a view to extracting from data information used to understand and predict the behavior of the biological system [150]. This is made possible by the extraordinary progress in molecular biology, particularly in genome sequencing, and the high-throughput measurements of the “omic” technologies. Systems biology approaches have been applied to vaccinology for scientific discovery and the prediction of immunogenicity or efficacy of vaccines [6]. The identification of molecular signatures that predict the immunogenicity of vaccines has several applications, such as the identification of novel correlates of immunity or protection, the identification of good or bad responders to vaccination, or the acceleration of vaccine development. The definition of functional signatures of immune responses based on cellular and molecular measurements can lead to the identification of correlates of immunogenicity. An example is the identification of molecular modules (patterns of gene expression) a few days after vaccination [151]. However, the computational analysis and interpretation of large-scale datasets are challenging. Modular transcriptional repertoires of co-dependent genes can be used to simplify the analysis and interpretation of the datasets [152]. The system-wide measurements of immune responses and the resulting modules of molecular signatures, once related to the results of vaccine trials, may lead not only to the identification of correlates of protection but also to the identification of protective immune mechanisms. The importance of the initial innate immune response or the stress response to orient the type of immune response, along with the importance of antibodies and the induction of Th1, Th2, Th17, and Treg cells, can be delineated through the statistical deconvolution of the data [152]. Although the approach is broadly applicable, the modular repertoires are system-specific [152]. Consequently, what has been defined by blood profiling, a convenient source of information, may not be transposable to other tissues, such as the MG.

The systems biology approach has been applied to the mastitis field. For example, transcriptomics has been used to identify genes that have key functions in the immune response to *E. coli* infection or peptidomics to characterize the inflammatory reaction to *S. uberis* infection [153,154]. An example relating to vaccination against *E. coli* mastitis is provided by the analysis of the transcriptomic and cytokinic analysis of blood, milk, and mammary tissue in the days following immunization and infection challenge [105]. Whole blood transcriptome analysis on day 7 after the booster injection of the antigen-distinguished cows that were immunized systemically from those that received the booster by the MG ductal route. The modules *Neutrophil degranulation*, *Chemokine signaling pathway*, and *TLR and inflammatory signaling* were characteristic of the locally immunized cows. Those cows fared better than the control unimmunized cows or animals from the systemic vaccination group upon intramammary challenge. After the challenge, the tissue-derived data pointed to the induction of local type 3 immunity by intramammary immunization. These results prompt the use of systems vaccinology in further mastitis vaccine studies.

#### 3.2.2. Improve Knowledge of Adjuvants for Ruminants to Guide the Immune Response

If we posit that the natural infection of the MG orients the host’s response towards a status quo, the ability to overcome this regulation will be a necessity to induce sterilizing immunity. One way to get this necessary redirection of the immune response is by using suitable adjuvants. Important modes of action of adjuvants are linked to the recruitment and activation of antigen-presenting cells (APCs) [155]. An effective way to activate T cells is through the presentation of antigens by DCs that undergo innate immune activation [156]. Activation by pattern-recognition receptor (PRR) ligands also orients the immune response, provided that the antigen and the PRR ligand are in the same endosome compartment, so that antigen processing and activation proceed in concert [157]. That is what occurs when APCs engulf whole bacteria, which provide both a glut of antigens and PRR agonists as built-in adjuvants. The downside of whole bacteria is that they impose a complex combination of antigens and immunomodulatory molecules. We have seen that the resulting immune response is not perfect. An illustration of this is the immune response induced by *S. aureus*. The peptidoglycan-associated lipopeptides and glycopolymers in the *S. aureus* cell wall modulate immunity by inducing an IL-10 response in monocytes and monocyte-derived macrophages, which results in weak IL-17 responses, whereas presentation by dendritic cells triggers a robust Th1/Th17 response [158]. Killed *S. aureus* elicits Th17 cells that secrete IL-17A and IL-10, whereas the same vaccination protocol with *Candida* elicits Th17 cells that produce IL-17A and IFN-γ in mice or humans [159]. Killed or live *S. aureus* do not elicit the same kind of immune response in cows [160]. If we aim to orientate the response, antigen choice and combination with a proper adjuvant are part of the answer. New generation adjuvants may be critical in steering the T cell response towards the proper balance of T cell types and profile of cytokine production [161]. Adjuvants are expected to modulate the strength, quality, and persistence of the adaptive immune response.

Improving our limited knowledge of adjuvants to shape cell-mediated immune responses in cattle is a necessity if we aim at developing cell-mediated protective vaccine responses. The effects of certain adjuvants may be species-specific. In humans, responses of proteins to aluminum adjuvant tend to be a mixture of Th1 and Th2 responses whereas in mice the polarization is towards a Th2 response [157]. In the bovine species, incomplete Freund adjuvant (IFA) induced antigen-specific neutrophilic inflammation in the MG as well as CFA, whereas CFA was necessary with guinea pigs [162]. Curdlan, a beta-1,3-glycan ligand of Dectin-1 and the complement receptor CR3, has been reported to bias the immune response towards the generation of Th17 and cytotoxic CD8 cells by acting on human dendritic cells [163]. However, Curdlan did not improve IFA when used in combination with Montanide adjuvant in cows [103]. A cationic adjuvant formulation (CAF01) based on trehalose dibehenate, which was proven very active in mice and humans, induces only weak immune responses in cattle [164]. Consequently, it will be important to verify the activity of adjuvants in ruminants, and even at the level of the species.

Adjuvants targeting different PRRs will orient the immune response differently. For example, poly-IC (a synthetic analog of double-stranded RNA), which is recognized through TLR3 and the cytosolic receptor MDA5 (melanoma differentiation-associated gene 5), stimulates Th1 cell immunity, including the generation of cytotoxic CD8 T lymphocytes in mice [157]. A low-toxicity derivative of LPS, monophosphoryl lipid A (MPLA), stimulates TLR4 by the TRIF pathway and biases the response towards Th1 immunity in mice and humans [165], and possibly in cattle [166]. The action of MPLA, which is used in the mastitis vaccine UBAC^®^ (Hipra S.A.), deserves to be delineated in detail in the bovine species. Agonists of C-type lectins co-expressed by dendritic cells are capable of regulating the balance between Th1 and Th17 cells in mice or human cell models [167,168], but their effects on the immune response of ruminants need confirmation.

Single-adjuvant vaccines often have limitations because they do not induce all the immune responses necessary for protection. A careful selection of adjuvants to be used in combination can result in complementary and even synergistic enhancement of immune responses to vaccines [169]. The selection needs detailed examination, often by empirical means, as the effect of combinations is often unpredictable. Systems vaccinology could be of great help to unravel the adjuvant combination effects.

#### 3.2.3. Make the Most of Vaccine Delivery

In order to induce local immunity, the route of immunization is a major determinant. Currently, all licensed mastitis vaccines are administered via the parenteral route. Systemic immunization is suitable for obtaining circulating antibodies. It is possible to elicit an antigen-specific T cell response in the MG using the parenteral route [103]. Nevertheless, the local intramammary route may be more appropriate than the parenteral route to induce immediate protection at the portal of entry of pathogens, as illustrated recently [105,117]. Several studies have established that the lactating MG reacts poorly to immunization but that the dry MG is a successful route to obtain circulating and local antibodies [21,170,171]. There is less information on T cell responses following intramammary immunization. The MG of ruminants is not considered as a mucosal organ, and the intestine-MG immune axis seems to be deficient [21]. However, there remains the possibility that mucosal immunity is induced by another route. Recent attempts of nasal immunization have shown that it is possible to induce a mammary response, either humoral or cell-mediated, to *S. aureus* antigens [172,173,174]. This prompts further studies, and gaining knowledge on lymphocyte homing to the MG would be of great benefit. Regional immunization (subcutaneous) in the area drained by superficial (supramammary) lymph nodes has also been evaluated, with divergent results when compared to the classical neck or dewlap injection sites [175,176]. These investigations deserve to be resumed with a focus on the evaluation of the mammary T cell immune response. In addition, determining the nature of the antigen-presenting cells and their functional phenotype in the mammary tissue remains an unfulfilled but necessary research task.

Delivery systems, such as water/oil emulsions, microparticles or nanoparticles, microcrystals (i.e., alum), immune-stimulating complexes (ISCOMS), liposomes, and virosomes, offer an array of possibilities in combination with antigens and immunomodulators. Emulsions and their depot effect are not necessarily helpful and may even be detrimental to T cell responses [140]. Emulsions and liposome formulations have been shown to induce different immune responses in cows but not in mice. Biodegradable particles offer versatile platforms by varying size, surface properties, and association with immunomodulatory molecules [177]. These options, as illustrated in Figure 3, indicate that the complexity and multiplicity of combinations, assayed through empirical and low-throughput processes, mean long delays before suitable combinations are found. Here again, correlates of protection and predictive biosignatures of vaccine safety and efficacy would be of great help. 

## 4. Conclusions

Past and current mastitis vaccines induce an unsatisfactory level of protection. We do not know how they operate, and we have only partial knowledge and understanding of the most useful immune defenses of the MG. Recent attempts to develop new vaccines are mainly empirical, and some appear promising. The design and development of mastitis vaccines are difficult. Difficulties stem from the peculiarities of the interactions of pathogens with the MG, which are conditioned by the idiosyncrasies of the MG immunobiology. The raison d’être of the MG is to secrete milk for the offspring’s survival. Milk is an abundant growth medium for many bacteria, which poses a threat to the MG in the event of bacterial colonization. However, the secretion of milk should be preserved, even during infection. This constrains the immune response and may explain why infection does not induce full protection. Another peculiarity of MG infections is that they do not require aggressive virulence factors from the pathogen, which only needs fitness attributes adapted to the MG niche. The lack of required virulence factors equates to a reduced number of potential protective antigen candidates for vaccine design. We can consider that the adaptation of the MG-associated pathogen has resulted from a long co-evolution of the bacteria with their host. Aside from the reduction of virulence, another adaptive trait is likely to be the capacity to bias the host immune response towards tolerance so that the immune response induced by infection favors the persistence of infection, albeit subclinical. Vaccination will have to elicit “unnatural immunity” by improving on the natural immunity induced by infection.

A number of solutions can be proposed to get out of the current mastitis vaccine deadlock. The root causes of vaccine shortcomings have not been taken into account in a conscious way. The preeminence of vaccinology over immunology; that is, focusing on the microbe and overlooking the host, is one of the main reasons. Overreliance on antibodies and relative neglect of the cell-mediated immune response, as well as our ignorance of the useful protective immune response, hence the lack of reliable correlate of protection, have ensued. Focusing on epithelial barrier defenses with a stress on type 3 immunity should lead to further breakthroughs. Another point of interest is the regulation of the immune response in the MG. We need to get a better understanding of the homing of T cells to the MG, the nature of local cell-mediated immunity, including the nature (phenotype and functions) of resident lymphocytes in mammary tissue, and how to induce this immunity. The definition of valid correlates of MG protection is necessary. Then, the research in vaccinology can be oriented towards the design of efficacious vaccines. This is not an easy task. The combination of adjuvant, vehicle, route of immunization, and the choice of antigens present many possibilities, most of which will not be satisfactory if we refer to past failures. The quest for efficacious mastitis vaccines is difficult, but we think that a critical assessment of the prevailing concepts and the introduction of new hypotheses will help us out of the mastitis vaccine deadlock. This would entail rekindling research on the basic immunology of the MG, which has been neglected in the last decade. Combining advances in immunology with the tools provided by modern vaccinology will offer new approaches to mastitis vaccine research and development. This prospect may incite pharmaceutical companies to reengage in the development of mastitis vaccines.

## Figures and Tables

**Figure 1 vaccines-10-00296-f001:**
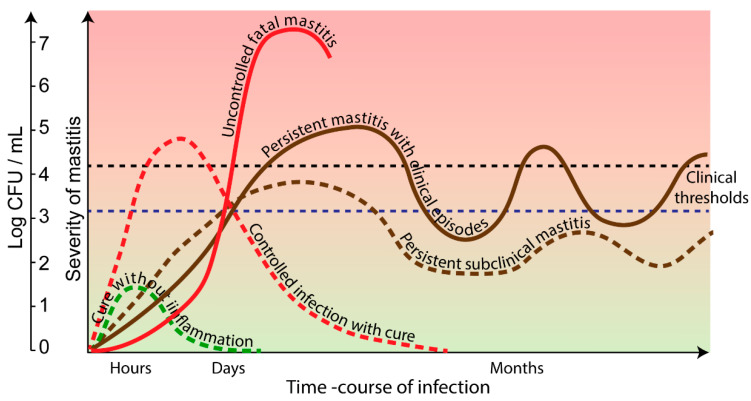
Different types of mastitis vaccination possible outcomes. The figure describes the time course of disease (bacterial cfu in milk) in unvaccinated (solid lines) and vaccinated (dotted lines) cows. In unvaccinated cows, mastitis may get out of control, usually after a sluggish immune response (red solid line), and the udder quarter is treated with an antimicrobial. In most cases, after a clinical episode, the infection persists in a subclinical state interspersed with short clinical episodes (brown solid line). In vaccinated cows, three schematic outcomes may occur. In the best but unfortunately unachieved case, the vaccine prevents infection (green dotted line). The coveted goal is to achieve sterilizing immunity, often after a short period of clinical mastitis (red dotted line). In most cases, the vaccine limits the severity of mastitis, maintaining the infection below the threshold of detection in a subclinical but persistent situation (brown dotted line). These outcomes are grossly linked to the concentrations of bacteria shed in milk, although variations occur as a function of the pathogen and the cow’s resistance or resilience. The clinical threshold is defined by the visible local and systemic symptoms, the lower threshold is by the milk appearance.

**Figure 2 vaccines-10-00296-f002:**
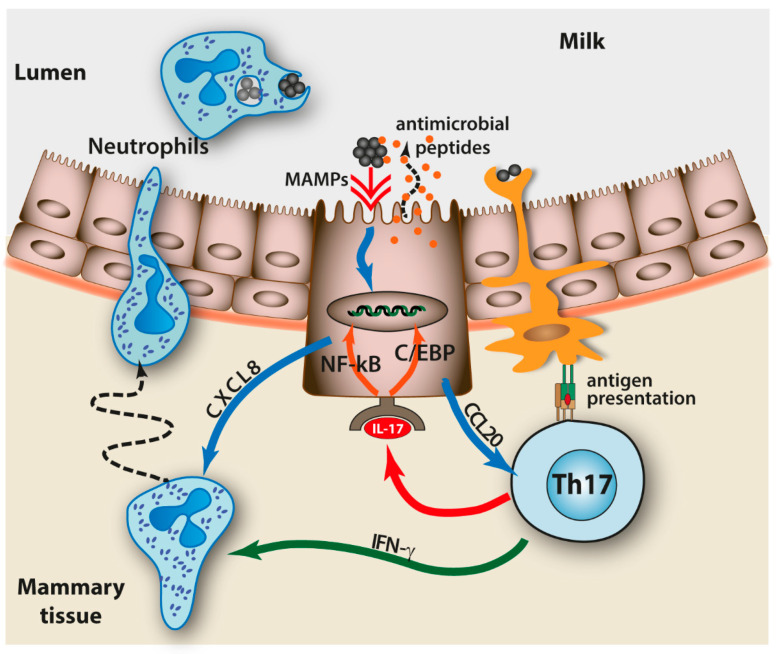
Mobilization of innate immune effectors by type 3 adaptive immunity in the bilayer epithelium of cisterns and large ducts. The capture of bacterial antigens and presentation by intra-epithelial antigen-presenting cells to local Th17 lymphocytes induces the production of IL-17 that activates mammary epithelial cells. In turn, these cells react to bacterial MAMPs and IL-17 by producing antimicrobial peptides at the luminal side and chemokines, such as CXCL8 and CCL20, at the basal side that attract neutrophils and other lymphocytes. The neutrophils, activated by the chemokines and IFN-γ, traverse the epithelium and reach the lumen where they can phagocytose invading bacteria.

**Figure 3 vaccines-10-00296-f003:**
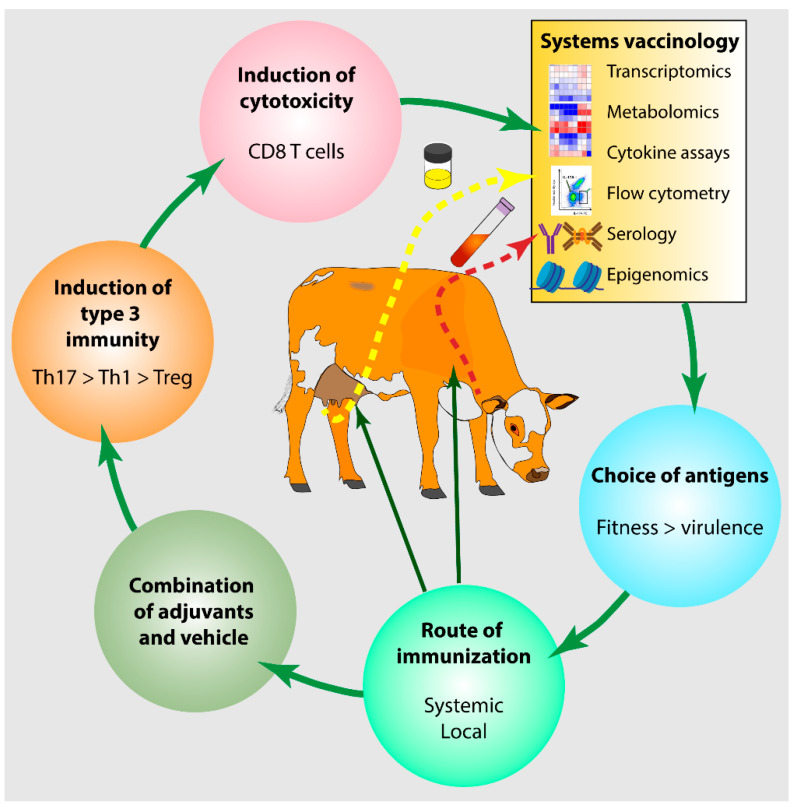
Possible levers to improve the effectiveness of mastitis vaccines. The tentative vaccine incorporates appropriate antigens related to bacterial fitness rather than virulence factors is administered preferably by systemic and local (intramammary) routes with a combination of adjuvants that orient the immune response towards type 3 immunity and the induction of CD8 T cells. The resulting immune response can be monitored with the instruments of systems vaccinology to instruct improvements in another cycle of vaccine development. See text for details.

**Table 1 vaccines-10-00296-t001:** Possible reasons for the current mastitis vaccine shortcomings.

Mammary Gland Features	Consequence for Defense Efficiency
Milk is a rich growth medium for many bacteria	Potentially high bacterial load
Dilution of antimicrobial defenses in milk	Reduced efficacy of antimicrobial agents
Absence of mucus barrier	Reduced efficiency of antibodies (sIgA) ^1^ and AMPs ^2^
Impediments to phagocytes (casein, fat globules, low oxygen tension)	Reduced phagocytic efficiency, need for massive leucocyte recruitment
Quenching of ROS ^3^ and AMPs ^1^ by milk	Blunting of antimicrobial activity
No need for specific virulence factors	No main target for the immune response other than fitness factors
Adaptation to the MG niche	Immune evasion
No protection following infection	Frequent recurrences

^1^ secretory IgA; ^2^ Antimicrobial peptides; ^3^ Reactive oxygen species.

**Table 2 vaccines-10-00296-t002:** Illustrative vaccine trials in cows against common mastitis pathogens.

Vaccine Antigens	Efficacy	Shortcomings	References
** *E. coli* ** **J5 bacterins**	Decreased severity of coliform mastitis in field experiments, little effect in experimental infections	Little effect on incidence of cases, variable among herds and experiments. Unknown mechanism	[80,81,82,83]
** *E. coli* ** **J5 bacterin with killed *S. aureus* (StartVac^®^, Hipra)**	Decreased mastitis severity in field studies	No effect on incidence of cases. Unknown mechanism	[84]
** *E. coli* ** **enterobactin FepA or siderophore receptor FecA**	Reduction of bacterial growth in vitro	Not tested in vivo (FepA) or not effective in challenge experiment (FecA)	[85,86]
** *Klebsiella* ** **siderophore receptors and porin proteins (KlebVax** **™** **)**	Little reduction in risk of coliform mastitis, some increase in milk yield	Administration with a J5 vaccine confounding the interpretation. Effect variable depending on experiments	[87,88]
** *S. aureus* ** **bacterins and toxoid or bacterial lysate**	Some reduction in severity and incidence of mastitis	Variable results, little prevention of chronic infections	[89,90]
** *S. aureus* ** **protein A**	Increased spontaneous cure after experimental challenge	Not tested in field conditions Mechanism not identified	[91]
** *S. aureus* ** **FnBP and ClfA**	Increased spontaneous cure after experimental challenge	Not tested in field conditions Mechanism not identified	[92]
** *S. uberis* ** **live bacteria and surface extract**	Reduction in bacterial shedding in milk and local inflammation	Not tested in field conditions Mechanism not identified	[93]
** *S. uberis* ** **SUAM**	Not reported	All cows developed mastitis. Not tested in field conditions	[94]
** *S. uberis* ** **slime preparation (UBAC** **^®^, Hipra)**	Reduction in milk production losses and incidence of clinical mastitis cases	Few published field experiments Little confirmed effect on the prevalence of infections. Mechanisms not identified	[95]

FepA, ferric enterobactin receptor; FecA, ferric citrate receptor; FnBP, fibronectin-binding protein; ClfA, clumping factor A; SUAM *S. uberis* adhesion molecule.

## Data Availability

Not applicable.

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
