# Peer review of "Progress towards the Elusive Mastitis Vaccines"

_vaccines, 2022, doi:10.3390/vaccines10020296_

Round 1

Reviewer 1 Report

1. This manuscript is a review article detailing the progress made in the field of mastitis vaccines. It explains the possible advantages and also the problems in establishing the immunity against the mastitis pathogens by available mastitis vaccines. Further, it details the immune mechanism of the mammary gland and their alveolar epithelial cells, type 3 immune mechanisms.

2. This review article is well organized and nicely presented. The topic is relevant in the field and also addresses the gap in the development of the mastitis vaccine against the bacterial pathogens causing the mastitis.

3. This article gives the overview of all the published articles in the field of mastitis vaccines and its immune mechanisms. It gives information about the problems and mechanisms for developing the mastitis vaccine for wider use and acceptability.

4. The conclusions are consistent with the evidence presented.

5. There is scope for improvement by adding more references in the field of immune mechanisms involved and the cytokine gene expression studies in experimental intramammary infection with major mastitis pathogens.

6. Tables and figures are adequate.

Further, the minor corrections are highlighted in the manuscript in the yellow colour and pop up notes with details in the PDF file.

Author Response

  1. This manuscript is a review article detailing the progress made in the field of mastitis vaccines. It explains the possible advantages and also the problems in establishing the immunity against the mastitis pathogens by available mastitis vaccines. Further, it details the immune mechanism of the mammary gland and their alveolar epithelial cells, type 3 immune mechanisms.
  2. This review article is well organized and nicely presented. The topic is relevant in the field and also addresses the gap in the development of the mastitis vaccine against the bacterial pathogens causing the mastitis.
  3. This article gives the overview of all the published articles in the field of mastitis vaccines and its immune mechanisms. It gives information about the problems and mechanisms for developing the mastitis vaccine for wider use and acceptability.
  4. The conclusions are consistent with the evidence presented.
  5. There is scope for improvement by adding more references in the field of immune mechanisms involved and the cytokine gene expression studies in experimental intramammary infection with major mastitis pathogens.

AU: Our review is a position paper and does not intend to analyze comprehensively the immune mechanisms involved in the defense of the MG against infections. We focused on the issues that we deemed the most relevant to the advancement of mastitis vaccine research, mainly with a speculative stance. We have just submitted a full review article of the adaptive immunity in the mammary gland of ruminants, which deals with immune mechanisms that can be mobilized by vaccination. The issue is vast, and cannot be covered in only one article. We understand that this can be a bit frustrating at the moment, but we hope that a satisfactory overview of the subject will be available soon.

  1. Tables and figures are adequate.

Further, the minor corrections are highlighted in the manuscript in the yellow color and pop up notes with details in the PDF file.

AU: x axis of Figure 1: the axis is to be taken as with a logarithmic scale, from hours, to days, to months. The early events take place in hours, whereas persistence may last for months. We added indications of duration units to the axis legend.

AU: Line 186: “patent” was replaced by “clear”.

AU: Line 444, a few references were added to support the production of type 3 immunity signature cytokines in infected mammary glands: Bruno et al. 2010; Blum et al. 2017; Andreotti et al. 2017. All the cited articles refer to ruminants, excluding the mastitis mouse model.

Reviewer 2 Report

The manuscript has been written by an authority in the topic of immunology of the mammary gland. It takes a innovative approach to presenting the topic and covers in general well and adequately.

I have two concerns, however.

First. I am not sure this paper should be called a review. It presents novel ideas, ok, based on previous literature references, but does not review them strictly speaking. It capitalises on them to make hypotheses and pathways. If I was the deciding editor, I would have redefined this submission as an original article.

Second. Section 3. Really, a table with details of vaccination trials with a brief comment on the efficacy and the possible shortcomings will enhance the final paper. This can be inserted also as a supplementary table to save a large re-arrangement of the manuscript.

Finally, a polish in English language is necessary. For example: effective, efficient, efficacious – I suggest to use consistently only one of these terms. There are a few more issues like this in the manuscript.

Author Response

The manuscript has been written by an authority in the topic of immunology of the mammary gland. It takes a innovative approach to presenting the topic and covers in general well and adequately.

I have two concerns, however.

First. I am not sure this paper should be called a review. It presents novel ideas, ok, based on previous literature references, but does not review them strictly speaking. It capitalises on them to make hypotheses and pathways. If I was the deciding editor, I would have redefined this submission as an original article.

AU: the Reviewer is correct in thinking that this paper is not a classic narrative review. It is more of a position or a perspective paper. As such, it does not seek to reference all relevant articles published in its scope, but selects those that support or discuss the points at issue. We have meant this manuscript as the complement of the essentially factual and classical review on mastitis vaccines recently published in the Journal of Dairy Science (Rainard, P., Gilbert, F.B., Germon, P., and Foucras, G. (2021). Invited review: a critical appraisal of mastitis vaccines for dairy cows. J Dairy Sci 104, 10427-10448), reference 2 of the present manuscript. We have taken care not to be redundant with this recent review.

Second. Section 3. Really, a table with details of vaccination trials with a brief comment on the efficacy and the possible shortcomings will enhance the final paper. This can be inserted also as a supplementary table to save a large re-arrangement of the manuscript.

AU: Two tables listing mastitis vaccine trials, detailing the targets, efficacy and pitfalls or knowledge gaps are included in the review mentioned above (JDS 104:10427). A sentence was added to mention this to the interested readers, on lines 36-37: “An overview listing the mastitis vaccine trials and their limitation is available in our previous review [2].

Finally, a polish in English language is necessary. For example: effective, efficient, efficacious – I suggest to use consistently only one of these terms. There are a few more issues like this in the manuscript.

AU: the terms effective, efficient, and efficacious share some meaning but have also some different connotations and senses. We deleted “efficient”, because according to the Concise Oxford Dictionary its first meaning is “productive with minimum waste or effort”, which is not appropriate in most sentences. We favored the use of “efficacious”, but sometimes retained “effective” where the intended meaning in the sentence was “having a definite or desired effect”. We have made changes throughout the text. Besides, a few sentences have been edited to improve readability.

Reviewer 3 Report

This is an interesting, thoughtful review of the current state of play in the development of vaccines for mastitis, incorporating a good discussion of the confounding aspects of the bovine mammary gland. Emphasis is placed on the importance of host (immunological) aspects that will accelerate vaccine development and improve the chances of success.

Minor comments:

Line 27: define MG on first usage.

Line 217: define SSC on first usage

Line 281: do you mean as compared to C. albicans? Note also that reference 61 does not describe cytokine responses in mice, it studies these responses in human cells, so this sentence / reference combination needs to be reconsidered.

Line 531: define MEC on first usage

Line 574: do you mean once applied, rather than once confronted?

Author Response

This is an interesting, thoughtful review of the current state of play in the development of vaccines for mastitis, incorporating a good discussion of the confounding aspects of the bovine mammary gland. Emphasis is placed on the importance of host (immunological) aspects that will accelerate vaccine development and improve the chances of success.

Minor comments:

Line 27: define MG on first usage.

Line 217: define SSC on first usage

AU: MG has been introduced on line 27 and “mammary gland” deleted on line 49. SCC has been defined on line 217.

Line 281: do you mean as compared to C. albicans? Note also that reference 61 does not describe cytokine responses in mice, it studies these responses in human cells, so this sentence / reference combination needs to be reconsidered.

AU: The sentence was edited to correctly refer to human cells, and make the comparison more explicit.

Line 531: define MEC on first usage

AU: MEC was introduced on line 237.

Line 574: do you mean once applied, rather than once confronted?

AU: What we mean is “set up against”, or “compared to” or “related to”. “confronted was replaced by “related to”.

Round 2

Reviewer 2 Report

The authors have made the recommended changes.
I only suggest that they add the table from the paper in JDS as supplementary material in the present review, at the same time providing details of the initial reference.

This will save interested readers to search for the article and will complement nicely with the present work.

Author Response

Response to Reviewer 2

The authors have made the recommended changes.
I only suggest that they add the table from the paper in JDS as supplementary material in the present review, at the same time providing details of the initial reference.

This will save interested readers to search for the article and will complement nicely with the present work.

AU: As recommended by the reviewer and to make the information accessible to all readers, we have added a table with details of vaccination trials as a main table, which is  referenced in the text (Line 369: illustrative examples in Table 2). We have removed the previous sentence that cited the reference where a similar table is available.